# Methodological Approaches to Experimental Evaluation of Neuroprotective Action of Potential Drugs

**DOI:** 10.3390/ijms251910475

**Published:** 2024-09-28

**Authors:** Igor Belenichev, Nina Bukhtiyarova, Victor Ryzhenko, Lyudmyla Makyeyeva, Oksana Morozova, Valentyn Oksenych, Oleksandr Kamyshnyi

**Affiliations:** 1Department of Pharmacology and Medical Formulation with Course of Normal Physiology, Zaporizhzhia State Medical and Pharmaceutical University, 69035 Zaporizhzhia, Ukraine; 2Department of Clinical Laboratory Diagnostics, Zaporizhzhia State Medical and Pharmaceutical University, 69035 Zaporizhzhia, Ukraine; 3Department of Medical and Pharmaceutical Informatics and Advanced Technologies, Zaporizhzhia State Medical and Pharmaceutical University, 69035 Zaporizhzhia, Ukraine; 4Department of Histology, Cytology and Embryology, Zaporizhzhia State Medical and Pharmaceutical University, 69035 Zaporizhzhia, Ukraine; 5Broegelmann Research Laboratory, Department of Clinical Science, University of Bergen, 5020 Bergen, Norway; 6Department of Microbiology, Virology, and Immunology, I. Horbachevsky Ternopil National Medical University, 46001 Ternopil, Ukraine

**Keywords:** ischemic stroke, models of cerebral ischemia, neuroprotection, drug effect

## Abstract

The authors propose a novel approach to a comprehensive evaluation of neuroprotective effects using both in vitro and in vivo methods. This approach allows for the initial screening of numerous newly synthesized chemical compounds and substances from plant and animal sources while saving animal life by reducing the number of animals used in research. In vitro techniques, including mitochondrial suspensions and neuronal cell cultures, enable the assessment of neuroprotective activity, which can be challenging in intact organisms. The preliminary methods help outline the neuroprotection mechanism depending on the neurodestruction agent. The authors have validated a model of acute cerebrovascular accident, which simulates key cerebrovascular phenomena such as reduced cerebral blood flow, energy deficit, glutamate–calcium excitotoxicity, oxidative stress, and early gene expression. A significant advantage of this model is its ability to reproduce the clinical picture of cerebral ischemia: impaired motor activity; signs of neurological deficits (paresis, paralysis, etc.); as well as disturbances in attention, learning, and memory. Crucial to this approach is the selection of biochemical, molecular, and cellular markers to evaluate nerve tissue damage and characterize potential neuroprotective agents. Additionally, a comprehensive set of molecular, biochemical, histological, and immunohistochemical methods is proposed for evaluating neuroprotective effects and underlying mechanisms of potential pharmaceutical compounds.

## 1. Introduction

The problem of ischemic stroke is currently becoming increasingly urgent. In industrially developed countries, stroke ranks 2–3 in the structure of overall mortality and is the main cause of permanent disability [1,2]. The trigger for ischemic neuronal death is energy deficiency, initiating the glutamate–calcium cascade—the release of excitatory aminoacidergic neurotransmitters—aspartate and glutamate and intracellular accumulation of Ca^2+^ ions. The processes that begin in the first hours of cerebral stroke and underlie the glutamate–calcium cascade (changes in glutamate and calcium metabolism, oxidative stress, hyperproduction of NO) induce remote consequences of ischemia—a genomic reaction with the inclusion of genetically programmed molecular mechanisms, dysfunction of the astrocytic and microglial pools, the development of immune changes, and the initiation of neuroapoptosis [3,4,5,6]. Despite certain successes achieved in the treatment of cerebral strokes, this problem still remains quite relevant. Modern neuroprotectors do not always show sufficient therapeutic effectiveness in clinical conditions, have a number of side effects with prolonged use, and due to the lack of a reliable therapeutic effect, they cannot be used in the acute period of stroke [7,8].

Currently, research is underway for newer neuroprotectors among various azaheterocyclic systems, natural compounds, neuropeptides, etc. [9,10,11,12]. The reliability and validity of the results obtained in the preclinical evaluation of potential drugs studied as promising correctors of cerebral blood supply disorders in cerebrovascular disorders are largely determined by how adequate the experimental models of cerebral ischemia are to the clinical manifestations of stroke [13,14] and, in particular, whether these models are capable of reproducing the main cerebrovascular phenomena: diffuse decrease in cerebral blood flow, energy deficiency, glutamate–calcium excitotoxicity, oxidative stress, expression of early response genes, etc. An important point in assessing the effectiveness of the cerebroprotective action of drugs is the ability of the experimental model to reproduce the clinical picture of cerebral ischemia: impaired motor activity; signs of neurological deficit (paresis, paralysis, etc.); impaired attention; learning and memory; lack of adequate orientation in space and time. The choice of biochemical, molecular, and cellular markers that allow us to assess the picture of damage to nervous tissue in this experiment and characterize the potential neuroprotector is also important to study.

In addition, in vitro studies for the neuroprotective evaluation of potential neuroprotectors under study are currently of particular relevance. These studies make it possible to study and establish the ability of a particular drug to influence a particular link in the pathogenesis of ischemic brain damage (oxidative nitrosative stress, glutamate excitotoxicity, shift in thiol–disulfide equilibrium, etc.).

## 2. Preliminary Evaluation of the Neuroprotective Effects of Potential Drugs In Vitro

Currently developed methods for obtaining individual nerve cells allow for high-quality and reliable information about the state of nervous tissue under pathological process modeling conditions and enable the study of the effects of potential neuroprotectors on them [15]. For in vitro studies, male white rats aged 4 weeks and weighing 80–100 g are used. Neuron and neuroglia enrichment fractions are isolated in two stages. In the first stage, brain tissue is disintegrated to obtain a cell suspension. In the second stage, differential ultracentrifugation is performed in a sucrose and Ficoll density gradient. The isolated neuronal cells are washed from the sucrose and albumin with a cold physiological solution (solution temperature 4 °C). Cell lines of nerve cells are also used [16]. 

The obtained suspension is divided into series as follows:-Intact: A suspension of neurons without the addition of initiating agents or potential neuroprotectors under investigation;-Control: A suspension of neurons to which agents that induce oxidative and nitrosative stress, glutamatergic excitotoxicity, and glutathione thiol–disulfide system deprivation are added at concentrations capable of causing the death of 50% of neurons (0.1–5 µM). To initiate oxidative stress in the neuron suspension, 0.25–1.0 mM H_2_O_2_ is added to the incubation medium. Glutamatergic “excitotoxicity” is induced by adding kainate (200–400 µM), glutamate (0.1–10 mM), or N-methyl-D-aspartate (100–150 µM) to the incubation medium. Glutathione thiol–disulfide system deprivation is achieved by introducing chloro-2,4-dinitrobenzene (CDNB) (50–500 µM), a selective inhibitor of glutathione S-transferase that forms conjugates with glutathione in cytosolic and mitochondrial fractions, into the incubation medium [16,17,18,19,20,21]. Alongside the intact and control samples, samples with the addition of initiating agents and pharmacological agents at various concentrations are prepared, followed by determining their effective concentration. The neuroprotective activity of potential neuroprotectors is assessed by counting neurons exhibiting signs of apoptosis using flow cytometry or histochemical methods. Our studies have shown that the addition of the aforementioned neurotoxins to the incubation medium led to a pronounced disruption of cellular, molecular–biochemical processes. These disruptions were consistent in nature but varied in degree of severity—accompanied by a sharp shift in the thiol–disulfide balance toward oxidized thiols (a significant decrease in reduced glutathione concentration and an increase in oxidized glutathione). An increase in the marker of oxidative protein damage—nitrotyrosine—was observed, along with a decrease in the activity of mitochondrial superoxide dismutase (Mn-SOD). We also recorded dynamic changes in the synthesis of endogenous cytoprotective factors—HSP and HIF proteins.

These pathobiochemical changes were detected as early as 15 min into incubation, reaching peak values by 60 min. Thus, in vitro studies have established that modeling glutamatergic excitotoxicity and depriving the glutathione component of the thiol–disulfide system led to the development of nitrative stress, resulting in irreversible molecular–biochemical and morphofunctional impairments of neuronal cells. It was found that the extent of these impairments correlated with changes in the expression of HSP and HIF proteins [8].

Recent research has established the central role of mitochondrial dysfunction in the processes of cell death under conditions of brain ischemia. In this context, it is pertinent to investigate the presence of what is referred to as “mitoprotective activity” in potential neuroprotectors [8].

For in vitro investigation of the “mitoprotective activity” of potential neuroprotectors, the mitochondrial fraction is isolated from rat brain tissue using differential centrifugation. In the incubation mixture (70 mM sucrose, 5 mM HEPES, 70 mM KCl, 0.5–1 mM KH_2_PO_4_, pH 7.4), a suspension of mitochondria (1 mg protein per sample) is added, along with the substances to be tested (10–100 µM in the sample), and incubated for 2 min. The opening of the mitochondrial pore in the mitochondrial suspension can be initiated by the following agents [22,23,24,25,26]:MPTP (1-methyl-4-phenyl-1,2,3,6-tetrahydropyridine)-induced opening of mitochondrial pores: add 40–60 µM MPTP to the incubation medium and after 5 min, add 50 µM CaCl_2_;Ca^2+^-induced opening of mitochondrial pores: add 200 µM CaCl_2_ to the incubation medium;NO-induced opening of mitochondrial pores: add 20–100 µM sodium nitroprusside to the incubation medium and, after 2 min, add 50 µM CaCl_2_;H_2_O_2_-induced opening of mitochondrial pores: add 50 mM hydrogen peroxide to the incubation medium and, after 2 min, add 50 µM CaCl_2_.

Mitoprotective activity is assessed based on the ability of the tested substance to prevent the opening of mitochondrial pores (MP) and reduce mitochondrial membrane potential (Ψ), as well as through a range of biochemical parameters [27].

We have determined that the addition of MPTP (50 µM) to the mitochondrial suspension led to pronounced disturbances in energy processes, characterized by a decrease in succinate dehydrogenase and malate dehydrogenase activities, along with an increase in nitrotyrosine, starting from the 30th minute of incubation. Additionally, mitochondrial dysfunction was observed, including MP opening and changes in mitochondrial membrane potential. The process of MP opening was most actively observed in the mitochondrial suspension at the 60-minute incubation mark [8]. It is important to emphasize that in vitro studies, both in neuronal suspensions and mitochondria, are useful for screening a large array of potential neuroprotectors, as well as for more detailed investigation of the most promising compounds to elucidate their mechanisms of neuroprotective action.

## 3. Preliminary Assessment of the Neuroprotective Effects of Potential Pharmaceutical Agents Using Various Models of Cerebral Ischemia

Despite potential drawbacks associated with experimental models of cerebral ischemia, these models are widely used in pharmacological, physiological, and pathomorphological research. This is primarily due to the relative ease of reproducing cerebral ischemia in animals and the ability to control biochemical, immunocytochemical, and histological changes in brain tissue at various time points after ischemia onset (seconds, minutes, days). These factors justify the use of brain ischemia models both for studying the mechanisms of ischemic neurodegeneration and for preclinical evaluation of potential neuroprotective agents [28,29,30].

Cerebral ischemia models can be reproduced in both large animals (such as rabbits, dogs, pigs, and primates) and small animals (such as mice and rats). Using large animals in studies of cerebral ischemia offers several advantages [31]. In large animals, it is easier to conduct complex physiological (such as electroencephalography, rheography, and gas exchange) and biochemical (including energy metabolism indicators, oxidative stress markers, and molecular markers of neural tissue damage) monitoring of the brain [32]. All these measurements can be performed simultaneously, at equivalent time intervals, on the same animals. However, modeling ischemia in large animals is a costly and labor-intensive endeavor, and experimental ischemia is not always reproducible. The use of large animals often requires various methods of anesthesia, which can also affect the quality of ischemia reproduction [33]. Additionally, the ethics committees and animal protection organizations do not support such research [34,35].

Using small animals has several advantages. Small animals, especially rodents, are less expensive. Mice, in particular, are genetically homogeneous, and genetic modifications can be relatively easily performed to create transgenic animals [36]. Another significant advantage, particularly with the use of mice and rats, is the ability to study complex behaviors, neuromotor functions, and memory to assess the severity of ischemia. The small size of the mouse and rat brains allows for procedures such as rapid freezing, which facilitates subsequent biochemical and cytochemical analyses [37,38]. The advantages and disadvantages of the most common models of ischemic stroke are represented in Figure 1.

There are several main experimental models of acute ischemic stroke (AIS), including global, focal, multifocal ischemia, and intracerebral hemorrhage. Modeling intracerebral hemorrhage involves either rupturing cerebral blood vessels or injecting autologous blood [14,39].

Most focal ischemia models involve occlusion of one of the major vessels supplying blood to a localized region of the brain, resulting in a reduction in local cerebral blood flow to 30%, which corresponds to the second critical level [31]. In multifocal ischemia, multiple distinct areas of reduced blood flow are observed. A characteristic feature of focal ischemia is the presence of a central ischemic necrotic core surrounded by a penumbra zone. This penumbra contains neurons that are damaged or dying but still viable, as well as neurons with no pronounced changes [40]. Most models of focal brain ischemia involve occlusion of the middle cerebral artery (MCA) in either small or large animals. MCA occlusion results in reduced cerebral blood flow in both the striatum and the cortex. The extent and distribution of ischemia depend on the duration of the artery occlusion, the location of the occlusion, and the amount of collateral blood flow to the MCA region [41,42].

There are various methods for artery occlusion to reproduce ischemia, either permanent or temporary, at proximal or distal sections of the vessel. These models have been extensively used due to their similarity to human stroke manifestations. The availability of these models has led to their widespread application for evaluating the effectiveness of different neuroprotective agents, understanding the mechanisms of brain damage during ischemia, and determining the roles of genes, proteins, lipids, and messengers involved in the development of ischemia [43,44]. A method for permanent proximal artery occlusions in rats has been developed, which includes subtemporal (semi-temporal) craniotomy. Using this model leads to the formation of an infarct zone in the cortex. In the permanent (constant) artery occlusion model, the local cerebral blood flow in cortical areas where histological anomalies occur is approximately 25 mL per 100 g per minute. The ischemic damage area corresponds to the region with reduced local cerebral blood flow [45,46,47]. A different variant of the focal brain ischemia model involves occlusion of the MCA in conjunction with occlusion of the common carotid artery [48]. Most researchers prefer using electrocoagulation for occluding the middle cerebral artery to reproduce ischemia [49]. 

Another model of focal ischemia involves thromboembolism of cerebral vessels in rats and larger animals. This model consists of introducing homologous blood clots directly into the common carotid artery or through a retrograde catheter placed in the external carotid artery [50]. Several models of brain ischemia–reperfusion have also been developed using a 30-minute temporary occlusion of the MCA. However, this technique requires advanced technical equipment [51,52]. There are recommendations for using this model of temporary occlusion in non-human primates, which provides a more objective assessment of behavioral, neurological, and cognitive impairments [53,54]. However, focal ischemia models induced by MCA occlusion have several limitations that restrict their widespread use in pharmacological experiments. Primarily, this technique is labor-intensive, requiring specialized equipment and involving the need for skull opening, which often damages the dura mater. This method does not facilitate neurochemical and biochemical studies. Only 30–45% of rats exhibit neurological deficits, and these are typically mild to moderate (up to 3–5 points on the McGraw scale). The recorded mortality of animals, particularly within the first day of this experiment, is usually due to complications associated with the reproduction of this model [8,14,41].

A relatively lesser-known method for reproducing focal permanent transient occlusion in rodents is the use of an intraluminal filament. This method has been widely used since its development in the late 1980s for studying both mechanisms of brain injury and neuroprotection. It involves two stages. In the first stage, the common carotid arteries are occluded, and in the second stage, an occluder is inserted intravascularly into the carotid artery and advanced to the middle cerebral artery. The occluder used is a chromic gut suture, which swells upon contact with blood and adheres firmly to the vessel walls. This ensures high reliability and reproducibility of this method. Due to its minimally invasive nature and the small area of ischemic damage, the mortality rate is 25–40% in animals. Animals that survive the procedure can be used to study post-ischemic recovery, including behavioral and exploratory activity, learning and memory, and mechanisms of neuroprotection [55,56,57].

A variant of focal ischemia models is photo-induced vessel thrombosis, where a constant ischemic focus is formed with respect to both volume and localization. This methodology is based on the principle of photochemical stimulation of thrombus formation in cerebral vessels through the interaction of a light beam with a fluorescent dye that has been previously introduced into the bloodstream. A limitation of this technique is that the photochemical reaction may induce microvascular damage. Focal photo-induced brain ischemia is primarily used in screening studies and behavioral research [58,59,60].

Global brain ischemia occurs due to disruption of blood flow in major arterial vessels (such as the aorta) or acute cardiac arrest. This model can be replicated through techniques such as carotid clamping or decapitation. It is characterized by selective damage to neurons located in various brain structures [61,62,63]. In global cerebral ischemia, the absence of cerebral blood flow leads to neuronal damage. Global brain ischemia can be classified into total (complete) and subtotal (incomplete) ischemia.

Total ischemia is utilized to study the regenerative capacities and delayed neuronal death in selectively vulnerable brain regions. The most straightforward method to induce complete global ischemia without recirculation is decapitation. This technique has been used for many years in small animals to investigate pathobiochemical changes associated with global ischemia. Following decapitation, the head is frozen and homogenized for metabolic and biochemical studies.

In experimental medicine, a cervical inflatable cuff has been used for an extended historical period to reproduce global ischemia in animals [64]. However, this method is associated with several complicating factors: venous congestion and compression of the vagus nerve, which can exert undesirable (distorting) effects on the development of ischemia. Cervical compression using an inflatable cuff has also been applied in large animals (dogs) to induce global brain ischemia. However, in this case, the vertebral arteries must be occluded separately due to their proximity to the spine. Cervical compression for 2 min using this method leads to loss of consciousness, followed by complete recovery. After 4–6 min of ischemia induced by this technique, animals remain in a comatose state for 24 h, after which there is a full recovery of normal neurological functions. After 8 min of compression, permanent neurological deficits occur. Similar procedures have been conducted in monkeys, involving a reduction in arterial pressure by 50 mm Hg before compression. Using this brain ischemia model in monkeys, hippocampal disturbances were observed after 15 min of ischemia, with neurological deficits persisting for many days following the ischemic event [65,66].

Ventricular fibrillation is another method for inducing complete global brain ischemia [67]. This technology is used to simulate the clinical condition of cardiac arrest, and many researchers further employ resuscitation to continue the state. This method has predominantly been used in large animals. Although it reproduces complete ischemia, it is labor-intensive and costly due to the subsequent need for animal resuscitation. In this method, cerebral perfusion pressure is low, and blood flow levels are significantly reduced compared to the control group despite attempts to increase perfusion pressure through the administration of adrenaline.

The cardiac arrest model conducted in mice (with 6–10 min of cardiac arrest) results in prolonged damage to the hippocampus and the sensorimotor cortex [68,69]. Ventricular fibrillation leads to ischemia in most organs. Therefore, models involving occlusion of the cephalic arteries in the neck and thoracic cavity are preferred, as they prevent complete brain ischemia while allowing for targeted ischemia in renal, visceral, and other areas [31]. A brain ischemia model has also been developed in cats, which involves occlusion of the ring-shaped and left subclavian arteries near their origins in the aortic arch, combined with a reduction in arterial pressure by administering ganglioblockers to levels below 80 mm Hg, and even as low as 50 mm Hg [8,70].

In this model, cerebral blood flow was reduced to near zero throughout almost the entire brain volume. As with other ischemic models, the consequences of pathological changes and behavior were proportional to the duration of the induced ischemia. Blood flow decreased to 10% of baseline levels in the neocortex, striatum, and hippocampus. Fifteen to thirty minutes after the removal of occluders, severe cerebral hyperemia developed, lasting up to 15 min. This was followed by acute cerebral hypoperfusion, which persisted for 24–36 h in the affected brain region. Currently, experimental models of complete global ischemia are rarely used in pharmacological research due to their complexity, high mortality rates, and the inability to perform prolonged dynamic monitoring of the animals [71].

Currently, there is significant interest in models of transient global brain ischemia induced by occlusion of major vessels. These models are used to study the mechanisms of delayed neuronal death in various brain regions and to identify potential neuroprotective agents [43,72].

In rats, a model of transient four-vessel occlusion is used. The modeling is performed in two stages. Initially, atraumatic occluders are placed near each common carotid artery and secured to the animal’s neck. The vertebral arteries are then electrocoagulated. After three days, the blood flow in the common carotid arteries is temporarily interrupted using the occluders [73,74]. This model promotes global ischemia: in 75% of animals, after 10–15 min of four-vessel occlusion, remote damage to pyramidal neurons in the CA1 and CA3 regions of the hippocampus was observed, while after 30 min of occlusion, neurons in the third and fifth layers of the sensorimotor cortex were affected. Mortality rates for this model range from 20% to 40% during the hyperperfusion period (1–3 days).

The four-vessel occlusion model is complex to perform and requires specific skills from the experimenter. 

Models of brain ischemia that induce moderate reductions in cerebral blood flow (subtotal ischemia) through irreversible occlusion of major vessels are more widely used in pharmacological research [14,41,75]. The most commonly used models involve unilateral or bilateral occlusion of the common carotid arteries, depending on the species-specific anatomical features of cerebral blood supply and collateral circulation capabilities in the animals. Mongolian gerbils are particularly preferred for this purpose due to their unique and convenient vascular anatomy, making them ideal for studying neuroprotection under brain ischemia conditions. A distinctive feature of gerbils is the disconnection of the circle of Willis in the cerebral circulation. To reproduce persistent global brain ischemia in gerbils, irreversible unilateral occlusion of the common carotid artery is performed. In this model, 30–40% of gerbils develop severe neurological deficits (15–20 points on the C. P. McGraw scale), while others experience moderate deficits and unilateral infarction 3–5 days after occlusion. The mortality rate in this model reaches 40–60% within the first 7 days post-occlusion [76,77,78].

In the absence of gerbils, white rats, both inbred and outbred strains, are commonly used to induce a similar level of ischemia. In these cases, ischemia is induced by bilateral occlusion of the common carotid arteries. This procedure results in a 50–60% reduction in cerebral blood flow, which gradually recovers to 85–90% and eventually returns to baseline levels within 2–3 days due to the compensatory activation of collateral circulation. The ischemic process involves sequential stages of brain perfusion disturbances: initial post-ischemic hyperemia is followed by a phase of post-ischemic hypoperfusion. Incomplete reperfusion limits the survival of ischemic tissue and leads to additional oxidative and osmotic brain damage. In 40–50% of the operated animals, severe neurological deficits develop (15–20 points on the C. P. McGraw scale), with persistent cognitive deficits observed up to 21 days post-occlusion. The mortality rate in this model reaches 60–80% within the first 7 days of observation.

To reduce mortality, a modified model involves complete, irreversible occlusion of one common carotid artery and a reduction in blood flow by 50–70% in the second common carotid artery. In this modified model, mortality is 40–60%. Bilateral ligation of the common carotid arteries is performed under general anesthesia through a surgical approach involving dissection of the carotid arteries and simultaneous placement of silk ligatures. It is important to note that this model is traumatic, and animal mortality, particularly in the first few hours, often results from errors made by experimenters, such as inadvertently ligating both the common carotid artery and the vagus nerve. Therefore, a properly executed procedure for modeling cerebral ischemia through bilateral occlusion of the common carotid arteries results in a mortality rate of 60–80%, depending on the strain of rats.

Bilateral occlusion of the common carotid arteries is associated with typical ischemia-induced biochemical disturbances in the brain: activation of glycolysis with hyperproduction of lactate, suppression of Krebs cycle enzymes and the electron transport chain, and ATP depletion accompanied by reduced expression of HSP70 and HIF-1α on days 1 and, at most, day 4 post-occlusion. By day 7 and, most notably, day 15 of this experiment, some improvement in energy metabolism in the brain was observed, but even by day 21, these metrics had not returned to control levels (sham-operated rats). This model of cerebral ischemia has demonstrated that inhibition of malate production and NAD-MDH activity correlates with decreased ATP, HSP70, and HIF-1α levels, as well as the severity of neurological impairments [8,79].

This model of cerebral ischemia is associated with persistent disruptions in the brain’s thiol–disulfide system, characterized by a reduction in the concentration of reduced glutathione, an increase in its oxidized form, and decreased expression of glutathione reductase and glutathione peroxidase 1 and 4. The most significant decrease in the activity of these enzymes on day 4 of this experiment correlates with diminished levels of glutathione and the overall thiol pool. This hypothesis is supported by two factors: firstly, the difference in the reduction in glutathione peroxidase (GPx) and glutathione reductase (GR) activity, particularly on day 1, and secondly, the maximal reduction in thiol-containing free amino acids in brain tissues on day 4 of this experiment [8,80]. The maximum accumulation of oxidative stress markers (such as nitrotyrosine, malondialdehyde, and carbonylated proteins) in brain tissues (CA1 hippocampus and cortex) occurred between days 1 and 4 of this experiment. Starting from day 15 of observation, the oxidative stress response was attenuated, but even on day 21, concentrations of nitrotyrosine and malondialdehyde (MDA) remained significantly higher than control levels (sham-operated rats). We found that on days 1 and, most notably, day 4 after bilateral occlusion of the common carotid arteries, there was an increase in the levels of Fe, Cu, and Mn in the cortex of the rats compared to the control group. The oxidation–reduction pair Fe^2+^/Fe^3+^ plays a crucial role in the activation of free radical reactions. Fe^2+^ is essential in all systems that generate reactive oxygen species, particularly in the formation of hydroxyl radicals through Fenton and Haber–Weiss reactions [8].

The disruption of the overall histostructure of the brain following bilateral irreversible occlusion of the common carotid arteries was characterized by pronounced perivascular and pericellular edema, significant vascular congestion, and accumulation of blood beneath the pia mater. There was also marked ischemic pathology in the ependymal cells of the brain ventricles. For the first 3 days after the bilateral occlusion, diffuse gliocyte hyperplasia predominated. Starting from day 5 post-occlusion, there was an increased intensity of nuclear staining in pyramidal neurons of the cortex and somewhat less so in the cytoplasm of cortical neurons.

Pathomorphological changes in the ependymal lining of the ventricles, such as the appearance of scalloped edges and desquamation of cells and cell fragments into the lumen of the lateral and third ventricles, were most pronounced between days 3 and 5 of this experiment. During days 1 to 3 of this experiment, there was an increase in nuclear size, with pyknotic nuclei observed on day 1. Changes in neuronal morphology were noted from day 1 through day 15, with the most significant alterations occurring on day 3 of observation. A decrease in the size of pyramidal neurons was evident on days 1 to 3, and by days 5 to 7, cell vacuolation in the tissue was observed, indicating brain edema. Morphometric studies revealed a reduction in neuronal density in the CA1 region of the hippocampus and layers IV–V of the cortex, decreased RNA concentration in the nuclei and cytoplasm, and increased density of apoptotic neurons, with the most pronounced changes occurring between days 1 and 5 of this experiment [8].

Among all the aforementioned experimental models of cerebral ischemia, only the subtotal ischemia model (bilateral irreversible occlusion of the common carotid arteries) is associated with pronounced pathobiochemical disturbances and activation of oxidative stress reactions in brain tissues. This makes it a suitable model for evaluating agents with antioxidant and anti-ischemic activity. Additionally, this model is preferred for studying cerebroprotective drugs that exhibit vasotropic, anti-edematous, antioxidant, and other protective activities. It allows for the assessment of the drug’s efficacy on brain regions with varying degrees of ischemia and the identification of potential side effects of vasotropic agents, such as the “steal” phenomenon.

The generalized methods for modeling AIS are presented in Figure 1, while the biochemical, cytochemical, and morphological consequences of AIS are detailed in Figure 2.

## 4. Assessment of Neurological Deficit in Animals with Experimental Cerebral Ischemia Serves as an Integrative Measure of the Neuroprotective Efficacy of Potential Pharmacological Agents

Neurological deficit in animals is typically assessed using the stroke-index scale developed by C.P. McGraw, which is widely used in experimental medicine [76].

The severity of the condition is determined based on the total score, with the following grading system:Mild: 0 to 3 points;Moderate: 3 to 7 points;Severe: 7 points and above.

Neurological deficits are assessed by observing symptoms such as paresis, paralysis of limbs, tremor, circling movements, ptosis, lateral positioning, and mobility. A specific test involves placing the rats on a rotating rod with a diameter of 15 cm and a speed of three revolutions per minute. The animals are tested daily, and scores are assigned as follows:Unilateral partial ptosis: 0.5 points;Unilateral ptosis: 1 point;Tremor: 0.5 points;Circling movements: 0.5 points;Paresis of limbs (per limb): 1 point;Paralysis of limbs (per limb): 2 points;Lateral positioning: 3 points;Inability to remain on the rotating rod (3 RPM) for 4 min: 3 points.

A registration card is created for each animal.

In addition, it is possible to evaluate the cognitive–mnestic abilities of an animal with a model pathology in various systems, such as a radial maze and a two-shuttle chamber, for developing a conditioned passive avoidance reaction.

## 5. Determination of Oxidative Stress Markers and Antioxidant System Status

Oxidative stress reactions, throughout their various stages (initial, free-radical, and peroxide stages), produce a range of products resulting from the interactions between free radicals and biological macromolecules. The levels of these products can indicate the intensity of oxidative stress within different regions of the brain, including neurons, glial cells, and subcellular structures. Therefore, these products can serve as markers of neurodestruction and the efficacy of potential neuroprotective agents [81,82]. The most important markers of oxidative stress are products of polyunsaturated fatty acid (PUFA) oxidation. These include short-chain alkanes and alkenes, as well as alkanals such as 2,4-alkadienals, alkatrienals, hydroxyalkenals, 4-hydroxyalkenals, and their peroxides, MDA, and normal aliphatic ketones. Isoprostanes, which are products of the interaction between arachidonic acid and free radicals, can also serve as markers of oxidative stress.

Markers of oxidative stress also include reactive oxygen species and stable metabolites of nitric oxide. The latter is formed from the metabolism of nitrite and peroxynitrite radicals, as well as from nitrosylated S-, N-, and O-macromolecules. Additionally, products of nucleic acid oxidation, such as 8-hydroxy-2′-deoxyguanosine, can be used as indicators of oxidative stress [83,84]. Each group of oxidative stress products reflects the intensity of oxidative stress and the degree of oxidative modification of lipids, amino acids, nucleic acids, and specific functional groups of proteins, including SH- and NH- groups, methionine SCH3 groups, and L-lysine amino groups. These modifications can lead not only to changes in proteins and enzyme activity but also to the destruction of bioantioxidants (such as vitamins, ubiquinone, steroid hormones, etc.), alterations in the phospholipid composition of biomembranes, and the appearance of oxidation products in the hydrophobic regions of the cell. These products can inhibit ion transport processes, alter the conformation of lipids and proteins, and consequently affect the structural and functional properties of membranes. However, the precise quantitative determination of oxidative stress products is an exceedingly complex process. This complexity arises primarily because, in biological systems, products of oxidative stress reactions are simultaneously metabolized (through oxidation, reduction, isomerization, polymerization, etc.). Additionally, a significant portion of these products is unstable and has a short lifespan. Furthermore, the low concentrations of these products necessitate highly sensitive physicochemical methods for identification. Moreover, the transformation of oxidative stress reaction products can occur during the processing of biological material, especially when examining ischemic tissues. For example, the free radical processes may be intensified when the homogenate or extracted system comes into contact with atmospheric oxygen. Therefore, the determination of oxidative stress products formed in ischemic tissues should be performed under conditions that prevent contact with air to avoid prolonging chain reactions [85,86].

Oxidative stress reaction products can be conditionally categorized into the following groups represented in Table 1 [8,87]:

The most widely used biomarkers for oxidative stress are stable products formed as a result of the oxidation of PUFAs, arachidonic acid, nucleic acids, tyrosine, and arginine. As previously noted, the peroxidation of PUFAs leads to the formation of peroxide compounds (hydroperoxides, endoperoxides, dialkylperoxides, and epoxides) and conjugated diene products. These conjugated dienes are formed through the redistribution of electronic density in the molecules of linoleic, linolenic, and arachidonic acids (but not oleic acid) [88]. Since diene conjugation appears during the formation of free radicals, its presence in lipid extracts from animal brains indicates the generation of free radicals and, thus, confirms the free radical mechanism of PUFA oxidation. 

Secondary products of oxidative stress arise from the breakdown of PUFA hydroperoxides, leading to the formation of a wide range of carbonyl compounds (n-alkenals, 2-alkenals, 2,4-alkadienals, alkatrienals, hydroxyalkenals, hydroperoxyalkenals, 4-hydroxyalkenals, 4-hydroperoxyalkenals, MDA, ketones, alkanes, and alkenes). Due to their chemical nature (stability), these compounds serve as primary markers of oxidative stress [89,90].

In addition to MDA, the primary carbonyl compounds formed from the peroxidation of ω-6 PUFAs include hexanal and 4-hydroxy-2,3-trans-nonenal; ω-3 PUFAs form propanal and 4-hydroxy-2,3-trans-hexenal. Beyond these compounds, free radical–peroxidative oxidation also produces 4-hydroxy-2,3-octenal, 4-hydroxydecenal, 4-hydroxyundecenal, 4,5-dihydroxydecenal, 4-hydroxy-2,5-nonanal, 2-hydroxyheptenal, 2-hydroxyhexenal, butanal, pentanal, octenal, and nonenal, though these are present in smaller amounts. Some of these products exhibit cytotoxic and mutagenic effects and can interact with biomolecules (such as proteins, nucleic acids, etc.), altering the structure of receptors, ion channels, and enzymes. They can also inhibit the synthesis of intracellular mediators and induce DNA and RNA damage. Unlike free radicals, carbonyl compounds are more stable and can exist both inside and outside the cell [91,92,93].

To determine secondary products, physical–chemical methods are commonly used. These methods are based on the absorption of energy by carbonyl compounds or their interaction products with analytical reagents in the ultraviolet region of the spectrum [94].

Among the stable products of oxidative stress reactions, MDA is commonly used as a marker. It is assessed through its reaction with thiobarbituric acid (TBA) and quantified spectrophotometrically [95].

This method is relatively simple to perform and has, thus, found widespread application. However, it lacks specificity because TBA reacts not only with MDA but also with other aldehydes, amino acids, carbohydrates, and bilirubin. These limitations of the TBA test have led to the development of numerous modifications. These include selective extraction of the azomethine complex using n-butanol and performing the reaction in the presence of iron (II) salts, which leads to the breakdown of hydroperoxides, among other adjustments [96]. Currently, fundamentally new methods have been developed for the determination of MDA and carbonyl compounds (such as formaldehyde, acetaldehyde, acetone, and both saturated and unsaturated aldehydes). These methods involve the use of specific reagents, including the following:Pentafluorophenylhydrazine (PFPH);Methylhydrazine (MH);4-(2-phthalimidyl)benzohydrazine (FBH);2,4-Dinitrophenylhydrazine (DNPH);o-(2,3,4,5,6-Pentafluorobenzyl)hydroxylamine hydrochloride (PFBH);tert-Butyldimethylchlorosilane (BDMCS);N,O-Di-(trimethylsilyl)-trifluoroacetamide (DTSFA);2-Hydrazinobenzothiazole (HBT).

These methods involve the quantitative determination of reaction products using techniques such as Thin-Layer Chromatography (TLC), Gas Chromatography (GC), High-Performance Liquid Chromatography (HPLC), Ultraviolet–Visible (UV–Vis) Spectrophotometry, Infrared (IR) Spectroscopy, and Mass Spectrometry (MS) [97,98,99].

Recently, stable products of oxidative reactions that can be used as markers of oxidative stress include oxidative modification products of arachidonic acid, specifically isoprostanes. 8-epi-Isoprostane (8-epi-PGFa2) can be accurately quantified in the daily urine of experimental animals [100].

Cyclic endoperoxides (PgG2-PgH2), which are formed from arachidonic acid through the action of cyclooxygenase, are unstable products and, thus, are not used as markers. Arachidonic acid hydroperoxides, which are produced by lipoxygenase, are inactivated by glutathione peroxidase (GPx) or leukotriene-A-synthase, converting into leukotrienes. It should be noted that arachidonic acid hydroperoxides are formed in relatively low concentrations, making their use as markers of oxidative stress less feasible [101].

Oxidative modification of nucleic acids also results in products that serve as markers of oxidative stress. For instance, the damaging effects of hydroxyl radicals and singlet oxygen on nucleic acids lead to the formation of products such as 5-hydroxymethyluracil, 8-hydroxyadenine, thymidine glycol, and 8-hydroxyguanine. 8-Hydroxyguanine is chemically stable, making it a useful marker for oxidative damage to the genome. Recent studies have shown that the level of 8-hydroxyguanine is significantly elevated in urine in various neurodegenerative diseases and after experiencing an acute stroke. Additionally, it is detected in urine much earlier than these diseases are clinically diagnosed [102]. Another significant biomarker of genome damage is thymidine glycol. However, it has not found widespread use as a marker of oxidative stress due to the more complex methods required for its isolation and detection [103]. During the activation of oxidative stress reactions in body tissues, the sequential breakdown of lipid hydroperoxides and the cleavage of alkoxyl radicals produce lower hydrocarbons. These hydrocarbons are exhaled, and their concentration in exhaled air serves as a specific and highly sensitive indicator of the intensity of oxidative stress processes. Quantitative analysis of hydrocarbons in exhaled air is conducted using GC. The primary end product of oxidative modification of PUFAs is pentane, which constitutes approximately 90% of the total, while the remaining 10% is made up of heptane and hexane [104]. Since methods for determining lower hydrocarbons are non-invasive, these markers are recommended for dynamic monitoring in larger animals during chronic experiments, as well as to help preserve the animal’s life.

Schiff bases, as products of the interaction between carbonyl compounds and the amino groups of proteins, amino acids, and nucleic acids, are also extracted and identified as markers of oxidative stress [105].

## 6. Determination of Antioxidant Enzyme Activity

For a comprehensive assessment of the neuroprotective effects of potential therapeutic agents, it is recommended to measure the activity of cytosolic superoxide dismutase 1 (SOD1) or CuZn-SOD and mitochondrial superoxide dismutase 2 (SOD2) or Mn-SOD, using both biochemical and immunoassay methods [15,106,107].

## 7. α-Tocopherol

α-Tocopherol, the most prevalent antioxidant in nature, is a lipophilic molecule capable of inactivating free radicals directly within the hydrophobic layer of membranes, thereby preventing the progression of lipid peroxidation. There are eight types of tocopherols, but α-tocopherol is the most active. α-Tocopherol donates a hydrogen atom to lipid peroxide radicals (ROO), reducing them to hydroperoxides (ROOH) and, thus, halting the development of lipid peroxidation processes. The α-tocopherol radical formed as a result of this reaction is stable and does not participate in the chain reaction. Instead, the α-tocopherol radical directly interacts with lipid peroxide radicals, reducing them while itself being converted to a stable oxidized form, tocopherol quinone. Reduced levels of α-tocopherol and tocotrienols in the context of brain ischemia have been shown to lead to the activation of oxidative stress and neuroapoptosis. In both in vitro and in vivo ischemic models, α- and δ-tocopherols have demonstrated anti-inflammatory effects. Furthermore, in addition to its antioxidant properties, α-tocopherol has been shown to modulate intracellular calcium activation and glutamate receptors, as well as to have delayed effects related to gene expression regulation [108,109]. At present, the determination of alpha-tocopherol in the brain is performed after lipid extraction using HPLC and spectrofluorometric methods [110].

## 8. Determination of the Most Informative Indicators of the Thiol–Disulfide System

According to recent data, the state of the thiol–disulfide system under brain ischemia conditions is a determining factor in the development of a cascade of pathobiochemical reactions, mitochondrial dysfunction, and cell death. Consequently, an essential component in evaluating potential neuroprotectors is their ability to influence the thiol–disulfide balance under conditions of induced brain ischemic damage [111]. To comprehensively assess the neuroprotective effects of potential pharmaceutical agents, it is recommended to measure the concentration of reduced and oxidized glutathione in the brains of animals, as well as the expression levels of glutathione reductase and glutathione peroxidase 1–4 [8,112]. 

## 9. Indicators of the Nitric Oxide System in the Brain

Pharmacologists are highly interested in the role of the molecular messenger nitric oxide (NO), which acts as a multifunctional physiological regulator in the mechanisms of neurodegeneration and neuroprotection of potential pharmaceuticals. This system has been the subject of numerous review articles, including our own [113,114,115,116]. We have demonstrated the direct involvement of NO in neuronal destruction by administering selective inhibitors of neuronal and inducible nitric oxide synthase isoforms to animals with focal cerebral ischemia and by conducting experiments on animals with a genetic deficiency in the gene coding for iNOS. Data indicate an increase in NO concentration in the brains of animals with focal and global ischemia. NO levels rise within the first minutes of ischemia, reaching a peak between 1 and 3 days. NOS activity sharply increases in the ischemic core and penumbra, but it is not possible to definitively attribute this to a specific enzyme type. However, there is evidence that neuronal NOS generates NO, which exacerbates neuronal damage, while endothelial NOS improves blood flow in the ischemic penumbra. This supports the role of NO in neuronal damage and death and highlights the specificity of NOS isoforms.

Additionally, the type and stage of stroke must be considered. It has been shown that the early stages of ischemia are characterized by the predominance of constitutional calcium-dependent NOS expression driven by transmitter autocoids. Neuronal cell death, under conditions of NO hyperproduction, begins with the activation of phospholipases, hyperproduction of hydroxyl radicals, and modulation of NMDA receptor activity. In the delayed post-ischemic period, from 7 to 14 days in global ischemia and from 1 to 3 days in focal ischemia, NO hyperproduction is observed as a result of the activity of inducible NOS activated by glia, macrophages, and neutrophils.

The independence of the inducible form of NOS from calcium allows it to maintain high activity for extended periods. Expression of this form under hypoxia occurs after approximately 6 h, in contrast to the constitutional calcium-dependent NOS, which is linked to the later appearance of activated astrocytes, macroglia, and inflammatory cells. In focal ischemia, these NO-producing cells are located in the penumbra, while in global ischemia, they are found in structures most in need of oxygen.

Consequently, studying the mechanisms regulating NOS activity is promising for developing treatment strategies for acute cerebral ischemia. Descriptions of positive outcomes from inhibiting NOS hyperactivity with specific inhibitors suggest that such treatment reduces the progression of brain ischemia [8]. The multifaceted role of the nitric oxide system in neuronal cell neurodestructive processes underscores the necessity of studying the ability of potential pharmaceuticals to affect this system. It is recommended to assess the activity and expression of different NOS isoforms using biochemical, immunoenzymatic, and histoimmunochemical methods and immunoblotting. Additionally, it is advised to measure stable NO metabolites, its thiol complexes, and nitrotyrosine.

## 10. Assessment of Brain Energy Metabolism Indicators

Occlusion of the vessels supplying the brain is the initial link in a chain of adverse changes that lead to severe disturbances in neuronal metabolism and structural and functional alterations, often culminating in neuronal cell death. Acute or chronic ischemia of brain tissue triggers a cascade of pathobiochemical reactions that ultimately result in focal neurological deficits, discirculatory encephalopathy, or even patient death. The close interrelationship between disturbances in energy and plastic metabolism and their impact on the course and prognosis of the disease is often overlooked in treatment planning, where the focus of pathophysiological therapy is typically on restoring hemodynamics.

Recently, significant attention has been given to disturbances in energy metabolism and the potential for its correction. Many researchers believe that correcting brain energy metabolism, both during the acute phase of a stroke and during the recovery phase, is a powerful preventive factor against recurrent strokes, patient disability, and mortality [117,118]. In this context, evaluating the neuroprotective effects of potential pharmaceuticals should include measuring key indicators of brain energy metabolism—such as ATP, ADP, AMP, glucose, glycogen, and glucose-6-phosphate. It is also essential to determine intermediates of the Krebs cycle (malate, isocitrate), pyruvate, lactate, and malate. Considering that the predominant role of the succinate oxidase mechanism has been inferred from studies on isolated organs and tissue cultures under varying degrees of ischemia and hypoxia, it seems prudent to investigate the state of limiting factors in energy metabolism and compensatory metabolic shunts, as well as the mechanisms of their molecular regulation during brain ischemia.

Moreover, concurrent analysis of various metabolic processes along with the levels of heat shock proteins (HSP70) and hypoxia-inducible factor (HIF-1α) can provide insight into the direction and extent of changes in these processes [119]. In modern biochemistry, the bioluminescence method using the enzyme luciferase (with sensitivity up to 10^−14^ M) is widely used for the determination of adenine nucleotides [120]. Chromatographic separation methods (such as ion exchange, thin-layer, and high-performance liquid chromatography) combined with subsequent fluorescence, spectrophotometry, or electrochemistry are considered sensitive. For preclinical studies, a simple and sensitive method for determining adenine nucleotides using thin-layer chromatography can be recommended. Additionally, it is advisable to measure intermediates of the Krebs cycle, anaerobic glycolysis, and compensatory shunts—such as the malate–aspartate shunt and the GABA shunt.

## 11. Morphometry of Various Brain Structures

To investigate the activity of potential neuroprotectors, conducting morphometric and densitometric studies in the IV–V layers of the cortex and the CA1 region of the hippocampus is particularly informative. The following parameters are recommended for assessment:Density of Neurons, Glial Cells, Apoptotic, and Destructively Altered Neurons: Measured as the number of cells per 1 mm^2^ of tissue section;Cellular Composition: Determined as the percentage of neurons, glial cells, apoptotic, and destructively altered neurons in the IV–V layers of the cortex and the CA1 region of the hippocampus;Area of Cell Bodies: Measured in µm^2^ for neurons, glial cells, apoptotic, and destructively altered neurons;RNA Concentration: In neurons, glial cells and apoptotic and destructively altered neurons are expressed in optical density units (ODU). This is calculated as the logarithm of the ratio of the optical density of the cell body to the optical density of the extracellular matrix;RNA Content: In neurons, glial cells and apoptotic and destructively altered neurons are expressed in ODU. This is calculated as the product of RNA concentration and cell area;Neuron Survival Index: Assessed as the ratio of the number of neurons in experimental animals to the number of neurons in intact control animals.

### 11.1. Markers with Informational Value in CNS Pathology

An important concept that has developed over the last decade is the idea of a “neurovascular unit” in neurodegenerative diseases. This unit encompasses blood vessels, nerve cells, and the extracellular matrix, all of which function together using various biochemical signals. Reduced blood flow in brain tissue leads to oxygen deficiency, initiating an ischemic cascade that includes calcium influx, neurotransmitter overstimulation, and increased production of free radicals. Ischemia triggers an inflammatory response in brain tissue, leading to biochemical disruptions that manifest as changes in the levels of various markers in the bloodstream.

In the early stages of inflammation, adhesion molecules (such as selectins and endothelial adhesion molecules) are involved. These molecules are expressed by endothelial cells and bind to glycoprotein receptors on the surface of neutrophils [121]. Activated microglia, macrophages, and leukocytes, in conjunction with neurons and astrocytes, release inflammatory mediators such as nitric oxide synthase, cyclooxygenase-2, IL-1, and monocyte chemoattractant protein-1. Due to the dual nature of activated microglia products—both destructive (free radicals) and protective (growth factors)—the role of microglia in cerebral ischemia is complex.

Transient activation of gene-encoding transcription factors (such as c-fos and c-jun) occurs within the first few minutes of stroke onset and triggers a second wave of expression of heat shock proteins, particularly HSP70. This expression peaks within the first 1–2 h of the stroke and then diminishes over the next 1–2 days. During the 6–24 h following the onset of stroke, there is a release of IL-1, IL-6, IL-8, and TNF-α. The levels of brain-derived neurotrophic factor (BDNF) and ciliary neurotrophic factor (CNTF) from oligodendrocytes increase.

Vascular endothelial growth factor (VEGF) exacerbates brain tissue edema during the acute phase of the stroke and is involved in vascular remodeling in the later stages. Besides local immune system imbalance, stroke primarily affects the neurovascular unit. An important factor is the disruption of basal membranes, which maintain neurovascular homeostasis. Markers associated with the ischemic cascade and subsequent inflammatory response are strong prognostic factors. These markers correlate with signs of early neurological impairment, the extent of damage, and early and late clinical outcomes [122,123,124]. Below is the list of a characterization of the most informative markers of cerebral ischemia:

#### 11.1.1. Gold Dot (NR2 Antibody Detection)

Excessive glutamate secretion caused by cerebral ischemia leads to hyperactivation of NMDA receptors. Excess NMDA receptors, particularly the NR2 subunit, are cleaved by serine proteases, resulting in the formation of NR2 peptide fragments.

Effectiveness of NR2 Antibody Detection:Independent Serum Marker: The level of antibodies to NR2 serves as an independent serum marker for cerebral ischemic events;Neurotoxicity Marker: NR2 antibodies are indicative of neurotoxicity associated with ischemic damage;Monitoring Tool: Tracking NR2 antibody levels enables monitoring the efficacy of pharmacological interventions for ischemic brain injury.

#### 11.1.2. Neuron-Specific Enolase (NSE) (Enzyme-Linked Immunosorbent Assay, Western Blot)

NSE is a neuron-specific marker. It is classified as an intracellular enzyme of the CNS, which enables the use of NSE for detecting post-ischemic brain damage.

#### 11.1.3. Myelin Basic Protein (MBP) (Enzyme-Linked Immunosorbent Assay, Western Blot)

MBP is released upon any damage to nervous tissue. Additionally, MBP levels increase over several days following ischemic modeling, reflecting the destruction of myelin sheaths.

#### 11.1.4. S-100 Protein (Enzyme-Linked Immunosorbent Assay)

S-100 is a specific protein of astrocytic glia that can bind calcium and is present in high concentrations in nervous tissue. An increase in the concentration of S-100 (αβ) and S-100 (ββ) in plasma serves as a marker of brain damage. The measurement of S-100B levels reflects the extent of brain injury. S-100 studies are useful for both monitoring and predicting the progression of pathology. Elevated S-100 levels in serum during cerebrovascular disorders are associated with microglial activation. Research has shown that in the early stages of ischemic brain injury, microglial cells in the peri-infarct area express S-100 and proliferate actively, with protein expression occurring for no more than three days post-ischemia. This suggests that the activation of resident microglia is an early response of brain tissue to ischemia and can be used as an early marker of damage.

#### 11.1.5. Galanin (Immunohistochemistry Enzyme-Linked Immunosorbent Assay, Western Blot)

Galanin is a 30-amino acid peptide that functions as a neuropeptide and a marker of functional activity in nervous tissue. It is synthesized in both the central and peripheral nervous systems. Galanin inhibits neurotransmitter release from neurons and plays a trophic role by enhancing neuronal survival following injury.

#### 11.1.6. Phosphorylated Neurofilament H (pNF-H) (Enzyme-Linked Immunosorbent Assay, Western Blot) 

It is a sensitive marker for axonal damage. Neurofilaments constitute the major part of the neuronal cytoskeleton. The three main neurofilament proteins are NF-L, NF-M, and NF-H. Their concentration is particularly high in axons. The NF-H protein has some unique properties. In axonal neurofilaments, serine residues of this protein, which are part of lysine–serine–proline repeats, are heavily phosphorylated. These forms of NF-H (pNF-H) are resistant to protease activity after being released from damaged axons. Consequently, the detection of this protein in plasma can provide information about the extent of axonal damage.

#### 11.1.7. Glial Fibrillary Acidic Protein (GFAP) (Enzyme-Linked Immunosorbent Assay)

GFAP is a member of the cytoskeletal protein family and constitutes the main 8–9 nm intermediate filament in mature CNS astrocytes. It is a highly specific brain protein that is not detected outside the CNS. GFAP is rapidly released into the blood following ischemic brain injury and can serve as a marker of injury severity and a prognostic factor. In the CNS, astrocytes respond to damage with astrogliosis, characterized by a rapid synthesis of GFAP. Due to its high specificity and early release from the CNS after ischemic brain damage, GFAP can be a very useful early marker of tissue destruction.

### 11.2. Markers of Neuroplasticity

#### 11.2.1. Neurotrophin-3 (NT3) and Neurotrophin-4/5 (NT4/5) (Immunohistochemistry, Enzyme-Linked Immunosorbent Assay, Western Blot)

The neurotrophin family includes nerve growth factor (NGF), brain-derived neurotrophic factor (BDNF), NT3, and NT4/5. These proteins support various neuronal populations. Neurotrophins (NTs) are secreted proteins detectable in the bloodstream that signal individual cells to promote survival, differentiation, or growth. NTs act by preventing the initiation of apoptosis in neurons. They also induce the differentiation of precursor cells and the formation of new neurons. NTs play a crucial role in the functioning of the nervous system and the regeneration of damaged neuronal structures.

Although the majority of neurons in the mammalian brain are formed during embryonic development, the adult brain partially retains the ability for neurogenesis—the formation of new neurons from neural stem cells. NTs regulate and stimulate this process. The trophic (supporting survival) and tropic (guiding axon growth) properties of NTs provide a basis for their potential use in treating various neurodegenerative diseases, such as Alzheimer’s, Parkinson’s, and Huntington’s diseases, as well as peripheral neuropathies of various origins.

NT3 is a growth factor with a molecular weight of 13.6 kDa (the molecular weight of the active dimer form is 27.2 kDa). NT3 plays a role in the development of the sympathetic nervous system. Elevated levels of NT3 have been detected in sympathetic ganglia and organs in mice with hyperinnervation and spontaneous hypertension. NT3 is capable of stimulating a broad range of neuronal populations, as it activates two out of three neurotrophin tyrosine kinase receptors (TrkC and TrkB). NT4/5 prevents the death of motor neurons during the perinatal and postnatal periods. The effects of NT4/5 are primarily mediated through the TrkB tyrosine kinase receptor [125].

#### 11.2.2. Expression of c-fos and c-jun Proteins in the Brain (Immunohistochemistry; Western Blot)

C-fos and c-jun proteins are classified as early response genes and are activated within the first hours of ischemia. The hyperexpression of these genes represents one of the earliest genomic responses to stress. Both c-fos and c-jun participate directly and indirectly in the fragmentation of DNA and the initiation of apoptotic cell death processes. There is a direct correlation between the expression of c-fos and c-jun genes and the induction of apoptosis mechanisms in neuronal cells. This correlation is mediated through the influence of these transcription factors on the expression of iNOS, leading to the overproduction of NO. NO, in turn, stimulates apoptosis by activating factors such as p53, Bax, caspases, and JNK kinase.

Additionally, c-fos was one of the first genes for which the involvement of its product in transcriptional regulation was demonstrated. This nuclear gene serves as a key nuclear target for signaling pathways that regulate cell growth and transformation and is involved in numerous cellular functions, including cell proliferation and differentiation. The assessment of these transcription factors allows for the prediction of disease progression and the evaluation of potential neuroprotective agents [126,127].

#### 11.2.3. Expression/Concentration of Heat Shock Proteins in the Brain (Immunohistochemistry, Western Blot, Enzyme-Linked Immunosorbent Assay)

In evolutionary terms, heat shock proteins (Hsp) are highly conserved and are found in all organisms, from bacteria to humans. This indicates that they perform fundamental cellular functions. The cytoprotective properties of stress proteins and their roles in normal cellular activities are largely due to their function as chaperones. Chaperones are proteins that facilitate the formation of secondary and tertiary structures of other proteins. Hsps also play a role in the repair or elimination of misfolded or denatured proteins.

According to contemporary classification, there are seven types of small heat shock proteins (sHsp), with some recent discussions mentioning up to eight types. These are classified either by molecular weight or by their cellular functions. Small Hsps include those with molecular weights of 25/27 kDa, 22 kDa, and 20 kDa, as well as high molecular weight Hsps such as Hsp110, Hsp100, Hsp90, Hsp70, Hsp60, and Hsp40.

Heat shock proteins are categorized based on their synthesis characteristics into constitutive and inducible types. Constitutive Hsps are synthesized continuously in the cell, and their production is not increased in response to stress. In contrast, the synthesis of inducible Hsps begins shortly after the cell is exposed to a damaging agent [119].

Inducible Hsp70 (immunohistochemistry, immunoblotting, enzyme-linked immunosorbent assay) is a protein whose expression is activated under stress conditions affecting the cell or organism. Hsp70 is essential for neuronal cell recovery, survival, and the maintenance of normal cellular functions. It functions as a molecular chaperone, preventing protein aggregation and repairing damaged proteins in response to cellular stress caused by adverse environmental factors or ischemia. Currently, there is ongoing research to identify potential neuroprotective agents capable of enhancing Hsp70 expression to harness its neuroprotective properties for therapeutic purposes [119].

#### 11.2.4. Hypoxia-Inducible Factor 1α (HIF-1α) (Immunohistochemistry, Immunoblotting, Enzyme-Linked Immunosorbent Assay)

HIF-1α is a transcription factor initially identified as a regulator of erythropoietin expression. HIF-1α is considered a leading transcriptional regulator of mammalian genes responsible for the response to oxygen deprivation. It is activated in key physiological sites that regulate oxygen pathways, enabling rapid and appropriate responses to hypoxic stress [119].

#### 11.2.5. Brain-Derived Neurotrophic Factor (BDNF) (Immunohistochemistry, Enzyme-Linked Immunosorbent Assay, Western Blot)

The mature mammalian BDNF molecule has a molecular weight of 13 kDa and consists of 119 amino acid residues. The structural identity of BDNF across different mammalian species allows for the use of both enzyme-linked immunosorbent assay and immunohistochemistry for its detection in various animal models. BDNF exhibits significant functional activity. During development, it plays a crucial role in the differentiation of neurons, maturation, survival, and synapse formation. In the adult organism, the primary function of BDNF is neuroprotection, safeguarding brain neurons from ischemic damage and protecting motor neurons from axonal injury-induced death [128].

#### 11.2.6. Ciliary Neurotrophic Factor (CNTF) (Immunohistochemistry, Enzyme-Linked Immunosorbent Assay, Western Blot)

CNTF belongs to a limited family of neurotrophic cytokines that includes leukemia inhibitory factor (LIF) and oncostatin M (OSM). CNTF is considered a key factor for the differentiation of developing neurons and glial cells. It supports trophic maintenance and plays a role in protecting damaged or axotomized neurons. Interest in studying CNTF is driven by its ability to promote neuronal survival [129].

#### 11.2.7. Pigment Epithelium-Derived Factor (PEDF) (Immunohistochemistry, Western Blot)

PEDF is a glycoprotein with a molecular weight of approximately 50 kDa, which is known for its multiple biological functions. It serves as both a neuroprotective and neurotrophic factor, affecting various types of neurons. PEDF has been demonstrated to be a potent activator of neuronal differentiation in retinoblastoma cells. In birds and mice, it supports the survival and differentiation of developing spinal motor neurons, maintains the normal development of amphibian photoreceptor neurons, and regulates opsin expression in the absence of retinal pigment epithelium (RPE) cells. In rats, PEDF acts as a survival factor for cerebellar granule neurons, protecting them from apoptosis and glutamate toxicity. It also protects motor neurons and developing hippocampal neurons from glutamate-induced degeneration. Additionally, PEDF has been shown to protect retinal neurons from cell death induced by peroxide in cell cultures.

### 11.3. Markers of Apoptosis

#### 11.3.1. Annexin V (Immunohistochemistry, Western Blot)

Annexin V, also known as placental anticoagulant protein I (PAP I), belongs to a family of calcium-dependent phospholipid-binding proteins and is also a potential vascular anticoagulant protein. Biochemical changes during apoptosis include the translocation of phosphatidylserine (PS) from the inner leaflet to the outer surface of the plasma membrane. The presence of PS on the membrane surface can be observed from the early stages of apoptosis through to complete cell degradation. Annexin V binds with high affinity to exposed PS on the surface of apoptotic cells and inhibits the procoagulant and proinflammatory activities of dying cells [130].

#### 11.3.2. Caspase-3 (Immunohistochemistry, Western Blot, Enzyme-Linked Immunosorbent Assay)

Caspase-3 is an enzyme that cleaves substrates at the carboxyl side of aspartate residues. The active form of caspase-3 consists of two active sites and is made up of two large (~20 kDa) and two small (~10 kDa) subunits derived from two precursor polypeptides. Caspase-3 is proteolytically activated by other caspases. Together with caspases-8 and -9, it is a central component of the apoptosis pathways [131,132].

#### 11.3.3. Cathepsins (Immunohistochemistry, Western Blot, Enzyme-Linked Immunosorbent Assay)

Cathepsins are a group of proteases comprising at least 15 different proteins. Most of these proteases are found in the lysosomes of various cell types, where they are activated at low pH levels and are responsible for the degradation of protein molecules [131].

#### 11.3.4. Procathepsin B (Immunohistochemistry, Western Blot, Enzyme-Linked Immunosorbent Assay)

Procathepsin B consists of 339 amino acid residues: a signal peptide (1–17), a propeptide region (18–79), and the mature chain (80–333). The active form of cathepsin B can activate caspases and prorenin and inactivate the secretory leukocyte protease inhibitor (SLPI). Key biological functions of cathepsin B include the activation of apoptosis and the regulation of the angiotensin–renin system. Cathepsins B and L play a crucial role in the development of the central nervous system (CNS); for example, mice with a combined deficiency of these proteases exhibit neuronal deficits and brain atrophy and typically die at 2–4 weeks of age [131,133].

#### 11.3.5. DR5 (Death Receptor) (Immunohistochemistry, Western Blot)

“Death receptors” (DRs) contain cytoplasmic death domains (DDs) that, upon binding with a death ligand, recruit adaptor proteins containing both DD and death effector domains (DEDs). The interaction between the DED domains of the adaptor and procaspase leads to autoprocessing and activation of the caspase cascade [131,134].

#### 11.3.6. Bcl-2 Family (Immunohistochemistry, Western Blot)

The Bcl-2 family of cellular proteins comprises 17 members that exhibit a wide range of activities, from inhibiting to promoting apoptosis. This family is categorized into subfamilies that differ functionally and structurally:Anti-apoptotic Subfamily: includes close homologs such as Bcl-2, Bcl-XL, and Bcl-w, which inhibit apoptosis;Pro-apoptotic Subfamilies: include proteins such as Bax and members of the BH3-only group, which promote apoptosis.

Apoptosis is associated with various mitochondrial changes, including the release of cytochrome c into the cytoplasm. Bcl-2-related proteins regulate these changes by forming channels in the mitochondrial membrane through which cytochrome c escapes into the cytoplasm. Bcl-2 and Bcl-XL inhibit the release of cytochrome c, while Bax promotes it. Additionally, Bcl-2 can inhibit Bax’s ability to form channels. Furthermore, Bcl-2 and Bcl-XL can directly bind cytochrome c, displacing it from the apoptosome and thereby preventing caspase activation [131,135].

We propose a methodological approach to evaluating the neuroprotective effects of potential therapeutic agents, which consists of two stages: **in vitro** studies and a model of cerebrovascular impairment. The current relevance of **in vitro** studies for neuroprotective assessment of potential neuroprotectors is highlighted by their ability to:Perform Preliminary Screening**: Evaluate a large number of molecules while preserving animal lives;Investigate Mechanisms**: Examine the impact of potential neuroprotectors on specific aspects of the ischemic brain damage pathogenesis, such as oxidative nitrosative stress, glutamate excitotoxicity, and thiol–disulfide balance shifts.

For a more in-depth evaluation of the neuroprotective effects of new molecules, we have developed and validated a model of acute cerebrovascular impairment (irreversible occlusion of the common carotid arteries). This model replicates key cerebrovascular phenomena, including the following:Diffuse Decrease in Cerebral Blood Flow;Energy Deficit;Glutamate–Calcium Excitotoxicity;Oxidative Stress;Expression of Early Response Genes.

An important benefit of this model is its ability to reproduce the clinical picture of cerebral ischemia, including motor activity impairment, signs of neurological deficits (paresis, paralysis, etc.), and disturbances in attention, learning, and memory.

Choosing appropriate biochemical, molecular, and cellular markers is crucial for assessing neuronal tissue damage and characterizing the potential neuroprotective agent. We also propose a comprehensive set of molecular, biochemical, histological, and immunohistochemical methods to evaluate both the neuroprotective effects of the potential drug and the specific mechanisms underlying these effects.

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
