# Peer review of "Methodological Approaches to Experimental Evaluation of Neuroprotective Action of Potential Drugs"

_ijms, 2024, doi:10.3390/ijms251910475_

Round 1

Reviewer 1 Report

Comments and Suggestions for Authors

Comments: The review article entitled “Methodological approaches to experimental evaluation of neuroprotective action of potential drugs” proposed a novel approach for a comprehensive evaluation of neuroprotective effects using both in vitro and in vivo methods, and it can be utilized for initial screening of numerous neuroprotective agents. This review is interesting and timely, some questions need to be addressed before publication. I suggest to polish and check it carefully.

1. In abstract, “This approach allows for the initial screening of numerous newly synthesized chemical compounds and substances from plant and animal sources while preserving animal life”. I’m confused about the expression “preserving animal life”.

2. The pathological biomarkers between current model and related animal/human models should be compared to evaluate the accuracy of models.

3. The authors summarized and reviewed a methodological approaches to experimental evaluation of neuroprotective action of potential drugs, but it feels for readers like developing an approach, it’s easy to misunderstanding.

4. Suggest to chose a widely accepted neuroprotective agents related to the current mechanisms to verify the efficiency of model.

Author Response

We thank the reviewer for evaluating our manuscript and for constructive feedback. Point-by-point responses are below:

The review article entitled “Methodological approaches to experimental evaluation of neuroprotective action of potential drugs” proposed a novel approach for a comprehensive evaluation of neuroprotective effects using both in vitro and in vivo methods, and it can be utilized for initial screening of numerous neuroprotective agents. This review is interesting and timely, some questions need to be addressed before publication. I suggest to polish and check it carefully.

Q1. In abstract, “This approach allows for the initial screening of numerous newly synthesized chemical compounds and substances from plant and animal sources while preserving animal life”. I’m confused about the expression “preserving animal life”.

Answer1. The proposed in vitro primary screening methods do not require a large number of animals, or they may not use any at all. Thus, fewer animals are harmed. Probably “saving animal life” reflects this statement better. We have changed it in the article.

Q2. The pathological biomarkers between current model and related animal/human models should be compared to evaluate the accuracy of models.

Answer2. The molecular and biochemical markers of neurodestruction/neuroprotection, oxidative stress, and the nitric oxide system provide an objective assessment of both experimental model pathology and the clinical picture of brain strokes. This is reflected in monographs, reviews, and original articles, including our own. There are recommendations for the use of these markers in clinical laboratory diagnostics, e.g.,

*Belenichev, I. F., Cherniy, V. I., Nagornaya, E. A., Bukhtiyarova, N. V., & Kucherenko, V. I. (2015). Neuroprotection and neuroplasticity. Kiev: Logos510.;

*Belenichev, I.; Popazova, O.; Bukhtiyarova, N.; Savchenko, D.; Oksenych, V.; Kamyshnyi, O. Modulating Nitric Oxide: Implications for Cytotoxicity and Cytoprotection. Antioxidants202413, 504. https://doi.org/10.3390/antiox13050504;

*Kamenshchyk, A.; Belenichev, I.; Oksenych, V.; Kamyshnyi, O. Combined Pharmacological Modulation of Translational and Transcriptional Activity Signaling Pathways as a Promising Therapeutic Approach in Children with Myocardial Changes. Biomolecules 202414, 477. https://doi.org/10.3390/biom14040477;

*Belenichev I, Aliyeva O, Popazova O, Bukhtiyarova N. Molecular and biochemical mechanisms of diabetic encephalopathy. Acta Biochim Pol. 2023 Nov 22;70(4):751-760. doi: 10.18388/abp.2020_6953. PMID: 37991083.;

*Belenichev IF, Aliyeva OG, Popazova OO, Bukhtiyarova NV. Involvement of heat shock proteins HSP70 in the mechanisms of endogenous neuroprotection: the prospect of using HSP70 modulators. Front Cell Neurosci. 2023 Apr 17;17:1131683. doi: 10.3389/fncel.2023.1131683. PMID: 37138769; PMCID: PMC10150069.)

Q3. The authors summarized and reviewed a methodological approaches to experimental evaluation of neuroprotective action of potential drugs, but it feels for readers like developing an approach, it’s easy to misunderstanding.

Answer3. In this article, we present our perspective on a multi-step approach to investigating the neuroprotective effects of potential drugs, from in vitro to in vivo stages, on experimental models of ischemic stroke. This includes conducting evaluative biochemical, physiological, molecular, and genetic tests using established standardized methods. This approach allow for a more objective assessment of the neuroprotective effects of these medications.

Q4. Suggest to chose a widely accepted neuroprotective agents related to the current mechanisms to verify the efficiency of model.

Answer4. The adequacy of the proposed models to the clinical manifestations of brain strokes has allowed us to conduct preclinical research on drugs that are used in clinical practice or have received approval and are undergoing Phase 2 clinical trials. These include Thiocetam (available in Ukraine at Arterium, in Russia at Ozonpharm, and in Serbia at Balkan Pharmaceuticals) and Angiolin (Report "Preclinical Study of Specific Pharmacological Activity (Cardioprotective, Neuroprotective, Endothelial Protective) of Angiolin Tablets," edited by I.F. Belenichev, Zaporizhzhia, 2018, 234 pages). We will include this at the end of the text before the conclusion.

Reviewer 2 Report

Comments and Suggestions for Authors

In this manuscript, Belenichiev et al review the currently available in vitro and in vivo methods for evaluating potential neuroprotective drugs. The authors discuss in detail the various models available. They provide an account the usefulness of the ischemic models to measure then neuroprotective efficacy and oxidative stress markers. In addition, the authors review various indicators and morphometry of the brain. Furthermore, they highlight the identification and analyses of transcription and growth factors and inflammatory markers and methods of identifications in ischemia.

Overall,

This review is a comprehensive overview of the available methods and models available to neuroprotective drugs in ischemia, in addition to highlighting the factors that are currently studied.

Pros

-       The authors suggest an in vitro method to screen for screening and characterizing neuroprotectors for ischemia.

-       The authors investigate Mito protective activity and develop a protocol for its characterization.

-       They suggest all the necessary preliminary assessments to be considered when choosing an in vivo model and for biochemical characterizations.  

-       They include the methodological approaches to evaluate neuroprotective drugs, in addition to the factors that need to be weighed in.

Cons

-       Why is the currently mentioned in vitro method better than any of the currently available methods?

-       Which in vivo method do they use and suggest for better characterization of neuroprotective drugs?

-       The abstract is deceiving in that, the authors do not propose a novel approach but highlight and review what is currently available. There are few sentences in the beginning and towards the end that suggest they have developed and validated a model. But otherwise, this review suggests a detailed account of what is currently available in ischemic studies.

The current manuscript can serve as a starting point of any researcher looking to evaluate neuroprotective drugs, as the title suggests. The authors will have to reword their abstract to align with the title and content.

Author Response

We thank the reviewer for the evaluation of our manuscript and the feedback.

Reviewer's comments and cons

Why is the currently mentioned in vitro method better than any of the currently available methods?

Which in vivo method do they use and suggest for better characterization of neuroprotective drugs?

The abstract is deceiving in that, the authors do not propose a novel approach but highlight and review what is currently available. There are few sentences in the beginning and towards the end that suggest they have developed and validated a model. But otherwise, this review suggests a detailed account of what is currently available in ischemic studies.

The current manuscript can serve as a starting point of any researcher looking to evaluate neuroprotective drugs, as the title suggests. The authors will have to reword their abstract to align with the title and content.

Response to Reviewer's comments

Currently, a significant issue in the development of new neuroprotective drugs is that after promising preclinical studies, the potential drug often proves ineffective in clinical trials. This situation prompts researchers in pharmacology and physiology to reconsider their approaches to the preclinical evaluation of neuroprotective effects.

We are eager to present our vision of a multi-step approach to the preclinical investigation of potential neuroprotective drugs. This concept involves initial in vitro screenings, followed by studying candidate compounds in models of cerebral ischemia, and subsequently conducting evaluative biochemical, physiological, molecular, and genetic tests. This approach utilizes established standardized methods. The proposed in vitro methods using neuron cultures with various agents inducing neurodegeneration not only allow for the assessment of the neuroprotective effects of potential drugs but also, depending on the initiating agent, can suggest the mechanism of action for these drugs.

Overall, we expect that our approach will enable a more objective evaluation of the neuroprotective effects of potential drugs.

Round 2

Reviewer 1 Report

Comments and Suggestions for Authors

Accept

Reviewer 2 Report

Comments and Suggestions for Authors

The authors addressed the comments.